# Kinetics and Selectivity Study of Fischer–Tropsch Synthesis to C$_{5+}$ Hydrocarbons: A Review

**Zahra Teimouri [1], Nicolas Abatzoglou [2] and Ajay K. Dalai [1,*]**

[1] Catalysis and Chemical Reaction Engineering Laboratory, Department of Chemical and Biological Engineering, University of Saskatchewan, Saskatoon, SK S7N 5A9, Canada; zat047@usask.ca

[2] Department of Chemical and Biotechnological Engineering, Université de Sherbrooke, Sherbrooke, QC J1K 2R1, Canada; Nicolas.Abatzoglou@USherbrooke.ca

[*] Correspondence: ajay.dalai@usask.ca; Tel.: +1-306-966-4771; Fax: +1-306-966-4777

**Abstract:** Fischer–Tropsch synthesis (FTS) is considered as one of the non-oil-based alternatives for liquid fuel production. This gas-to-liquid (GTL) technology converts syngas to a wide range of hydrocarbons using metal (Fe and Co) unsupported and supported catalysts. Effective design of the catalyst plays a significant role in enhancing syngas conversion, selectivity towards C$_{5+}$ hydrocarbons, and decreasing selectivity towards methane. This work presents a review on catalyst design and the most employed support materials in FTS to synthesize heavier hydrocarbons. Furthermore, in this report, the recent achievements on mechanisms of this reaction will be discussed. Catalyst deactivation is one of the most important challenges during FTS, which will be covered in this work. The selectivity of FTS can be tuned by operational conditions, nature of the catalyst, support, and reactor configuration. The effects of all these parameters will be analyzed within this report. Moreover, zeolites can be employed as a support material of an FTS-based catalyst to direct synthesis of liquid fuels, and the specific character of zeolites will be elaborated further. Furthermore, this paper also includes a review of some of the most employed characterization techniques for Fe- and Co-based FTS catalysts. Kinetic study plays an important role in optimization and simulation of this industrial process. In this review, the recent developed reaction rate models are critically discussed.

**Keywords:** Fischer–Tropsch synthesis; syngas conversion; selectivity to C$_{5+}$; reaction mechanisms; catalyst deactivation; kinetics; characterization

## 1. Introduction

As a result of the increase in the world's population, there is a high demand for renewable and sustainable energy resources instead of fossil fuels. Therefore, alternatives such as Fischer–Tropsch synthesis (FTS), which is a promising route for clean fuel production, is gaining more importance in petroleum industry [1]. FTS has drawn much attention because of some factors: (a) increase in number of discovered natural gas fields, (b) environmental obligations for CO$_2$-neutral fuels production, (c) increase in efficiency of FTS in terms of catalyst design for liquid fuel synthesis, and (d) high price of escalation of crude-oil based technologies [2]. FTS technology consists of syngas production (H$_2$/CO mixture), converting the syngas to a broad spectrum of hydrocarbons (C1 to C100) by means of a heterogeneous catalyst and refining. Syngas is obtained by conversion of carbonaceous feedstocks such as biomass, coal, or natural gas to a mixture of CO and H2 by means of gasification and steam reforming [3]. The production and purification of syngas is considered the most expensive part of this technology [4]. So, the second step of this technology, which deals with the conversion of syngas to hydrocarbons, forms the heart of this process and has drawn considerable research interest. FTS is a catalytic reaction in which CO and H$_2$ molecules participate in a polymerization reaction by means of an active metal (Fe, Co, Ru and Ni). The reactants at first are converted to monomers and initiators, the monomers then polymerize to produce a variety of longer chain hydrocarbons such as

paraffins, olefins, wax and oxygenates. Possible reactions of the Fischer–Tropsch synthesis are summarized in Table 1.

**Table 1.** Significant reactions in Fischer–Tropsch synthesis (FTS) [5].

| Reaction Name | Related Equation |
|---|---|
| Paraffins formation | $nCO + (2n + 1)H_2 \rightarrow C_nH_{2n+2} + nH_2O$ |
| Olefins formation | $nCO + 2nH_2 \rightarrow C_nH_{2n} + nH_2O$ |
| Water-gas-shift reaction | $CO + H_2O \rightleftharpoons CO_2 + H_2$ |
| Alcohols formation | $nCO + 2nH_2 \rightarrow H(CH_2)_nOH + (n-1)H_2O$ |
| Boudouard reaction | $2CO \rightleftharpoons C + CO_2$ |

In FTS, a mixture of liquid and solid-like hydrocarbons ($C_{5+}$ hydrocarbons) is achieved as the final product, which is called Syncrude. It is noteworthy to add that the gaseous hydrocarbons ($C_1$–$C_4$) are mainly considered as byproducts of FTS. Syncrude contains a mixture of synthetic naphtha, synthetic middle oil distillates (diesel fuel and kerosene), lubricating oils, and synthetic waxes. Table 2 summarizes the characteristics and possible applications of these phases. Syncrude is recovered from the unreacted syngas by subsequent cooling and phase separation strategies such as distillation [6,7]. Recent studies have focused on tunning the selectivity of FT products to $C_{5+}$ hydrocarbons by changing catalyst structure, operating conditions, and reactor design. This review focuses on the recent achievements in the Fischer–Tropsch technology and its catalyst design, support materials, and kinetics study to increase the selectivity of $C_{5+}$ hydrocarbons.

**Table 2.** Characteristics and applications of FTS products.

| Product Name | Characteristic | Application |
|---|---|---|
| Synthetic naphtha | - Mixture of linear $C_{;5}$–$C_{;11}$ hydrocarbons<br>- Boiling point: 140–205 °C | - Raw material for ethylene and propylene production |
| Synthetic kerosene | - Linear $C_{;10}$–$C_{;14}$ hydrocarbons<br>- Boiling point: 150–180 °C | - Raw material for the manufacture of surface-active compounds<br>- Jet engine fuel |
| Synthetic diesel fuel | - Linear $C_{;11}$–$C_{;18}$ hydrocarbons<br>- Boiling point: 180–360 °C | - Transportation fuel |
| Lubricant oil | - $C_{18}$–$C_{44}$<br>- Boiling point: ~300 °C | - Lubricating oil for the reduction of friction, heat, and wear in motorized vehicles |
| Synthetic waxes | - $C_{20}$–$C_{60}$<br>- Boiling point: >360 °C | - Hot melt adhesives (HMA)<br>- Printing inks and coatings<br>- Bitumen modification<br>- Polymer processing<br>- Polishes and textiles |

## 2. Mechanisms of FTS

Over the past 90 years since the introduction of the Fischer–Tropsch process, researchers have proposed several mechanisms for formation of hydrocarbons on the surface of the catalyst. The complex product spectrum of this technology has prevented reaching a general agreement on the mechanism of FTS [8,9]. Understanding the underlying mechanism of the reaction plays a pivotal role in designing an effective catalyst for FTS. The suggested mechanisms differ in the structure of monomer and initiator species and are based on two hypotheses: (1) cleavage of C–O bonds, which is followed by $CH_x$ species formation, or (2) insertion of CO into the main chain, producing $RCH_xOH$ species [10]. Carbide, Enol and CO insertion models are among the proposed mechanisms for FTS, which are discussed in the following sections.

*2.1. Carbide Mechanism*

Carbide mechanism is one of the initial mechanisms that was first proposed by Franz Fischer and Hans Tropsch (1926). It proposes the formation of $CH_x$ intermediates by means of the hydrogenation of surface carbon that arises after the dissociation of adsorbed CO [11]. These intermediates are supposed to act as monomer of chain growth. Termination step of the polymerization reaction occurs by two routes: (a) desorption of the unsaturated intermediates to result in olefins or (b) addition of $CH_3$ species or hydrogen to yield paraffins [12]. Although there is large experimental support for the carbide mechanism, this mechanism is faced with limitations for explanation of branched isomers and oxygenated products such as alcohols and acids since a considerable amount of oxygenates are formed during FTS. Therefore, formation of these products cannot be ignored. On the other hand, it was recognized that the assumptions made in this mechanism are not in agreement with thermodynamic data of the hydrogenation of the carbide under FT reaction conditions [13]. By using thermodynamic data, Kummer et al. [14] have indicated that, the hydrogenation of iron carbide to form hydrocarbons is not the real reaction pathway. In another work, Kummer et al. [15] investigated the direct hydrogenation of carbide by the reaction of a reduced iron catalyst and radioactively labeled $^{14}$CO. They found that direct hydrogenation was responsible only for a small fraction of the methane production (8–30%). This mechanism is outlined in Figure 1. Experimental data indicated that $CH_2$ intermediate does not undergo self-polymerization and there are other species that act as monomer and initiator of the chain growth. These assumptions led to instruction of other modified mechanisms. Alkyl, alkenyl and, alkylidene–hydride–methylidyne mechanisms are based on the modified carbide mechanism. Alkyl mechanism, which was proposed by Brady and Pettit [16], assumes that chemisorbed $CH_2$ acts as monomer like in the original carbide model but chemisorbed $CH_3$ is responsible for chain initiation. On the other hand, Maitlis [17], with regard to the alkenyl mechanism, proposed that an adsorbed vinyl specie ($CH_2 = CH$) acts as chain initiator and the chemisorbed $CH_2$ is again in charge of chain growth monomer. In alkylidene-hydride-methylidyne, Ciobica et al. [18] investigated the possible reaction pathways for FTS over Ru by ab initio calculations. They suggested that instead of adsorbed $CH_2$, adsorbed CH + H is the chain-growth monomer, and an isomeric vinyl compound $CH–CH_2$ initiates the chain-growth. An alternative for carbide mechanism, which considers direct CO dissociation followed by hydrogenation to produce $CH_x$ species, involves H-assisted CO dissociation. Recently, Broos et al. [19] carried out a research based on quantum-chemical density functional theory (DFT) to study the mechanism of CO dissociation by using χ-$Fe_5C_2$ Hägg carbide. According to their results, the direct C–O bond dissociation was dominant on the stepped sites of Fe carbide, on the other hand, the H-assisted CO dissociation mechanism proceeded on the surface. In another research, Broos et al. [20] investigated the CO dissociation path by using low-index Miller planes of θ-$Fe_3C$ based on quantum-chemical study. Direct CO dissociation was more important for step-like sites on the θ-$Fe_3C$ surface by implying lower energy barriers. They also found that the high temperature θ-$Fe_3C$ phase is active for CO bond dissociation, and the barrier for dissociation was lower compared to those on α-Fe and χ-$Fe_5C_2$ phases. In another work, Helden et al. [21] investigated the CO dissociation route on the stepped fcc-Co (211) surface using the Vienna ab initio simulation package (VASP). They found that the overall barrier for CO dissociation via HCO intermediate formation was 123 kJ mol$^{-1}$, which was lower compared to that (142 kJ mol$^{-1}$) in the direct CO dissociation path. These studies provide strong experimental support for the H-assisted CO dissociation mechanism on Fe and Co catalysts in FTS.

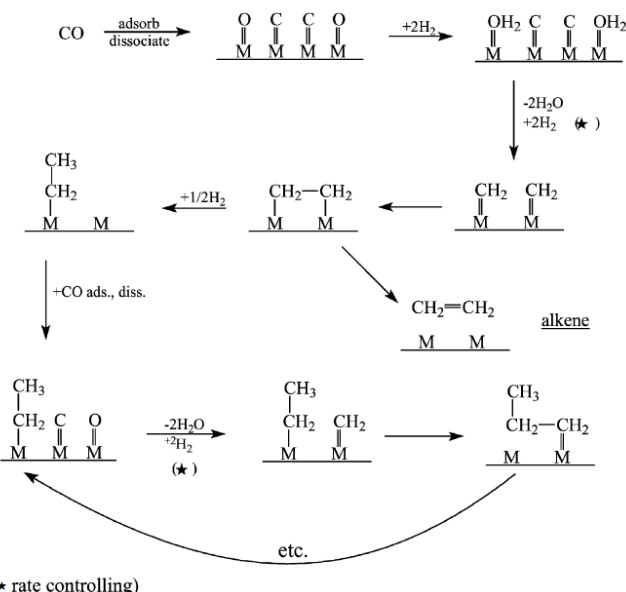

**Figure 1.** Schematic of carbide mechanism. Reproduced with permission [13]. ★: rate controling step.

*2.2. Enol (Oxygenate) Mechanism*

Considering the limitations of carbide mechanism, oxygenate (enol) mechanism was proposed (the 1950s) and gained widespread acceptance. This mechanism postulates the non-dissociative adsorption of CO, which is then hydrogenated to form an enol compound (HCOH). Enol compound grows by using adjacent groups as a result of the condensation-water elimination reactions [13]. The formation of branched hydrocarbons is due to the presence of a CHROH surface species. Experimental works done by Kummer and Emmett [22] and Kummer et al. [23] have provided a strong support for this mechanism. They introduced a [14]C-labeled alcohol or alkene simultaneously with syngas as feed of the reactor and analyzed the distribution of products. According to their investigation, the added labeled alcohol or alkene was able to start the chain growth. The schematic of this mechanism is shown in Figure 2.

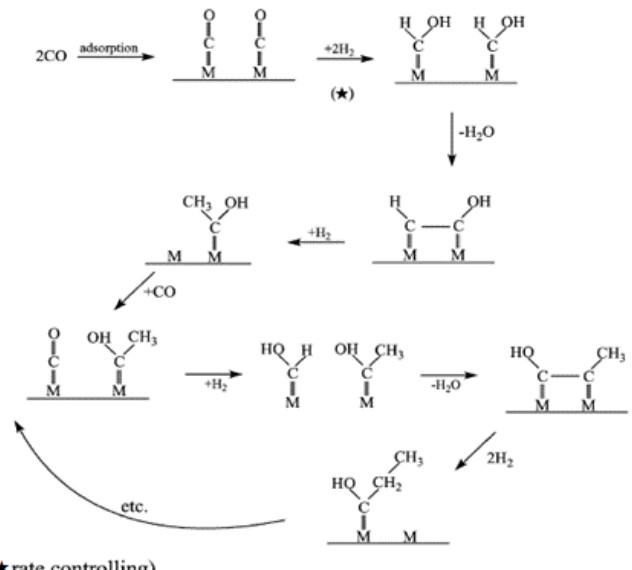

**Figure 2.** Schematic of enol mechanism. Reproduced with permission [13]. ★: rate controling step.

*2.3. CO Insertion Mechanism*

Although the enol mechanism successfully explains the formation of alcohols, the other mechanism, which is called CO insertion, offers a simpler route to this end. This model was introduced by Pichler and Schulz [24] and explains that adsorbed carbon monoxide on the surface of the catalyst acts as a reaction monomer, and chain growth propagates via insertion of an intermediary carbonyl at the metal-alkyl or metal-hydride bond. This insertion results in an acyl group, which is subsequently hydrogenated to produce an alkyl group with an additional methylene group and water as the byproduct. Micro-kinetic models, isotope tracer studies, and steady state and transient kinetic investigations provide strong support for this mechanistic scheme, where C–O scission is the key step [25–29]. Recently, Zhang et al. [30] conducted research to elucidate the reaction pathways over Co catalyst in FTS. Their results indicated that there was no reaction between CO and $H_2$ at low temperatures (140 °C) but CO reacted in the presence of $C_2H_4$ simultaneously with syngas. This observation proposed that the adsorbed CO and $C_2H_4$ act as monomer and initiator, respectively. In other words, their results indicated that C–C bond coupling precedes the C–O bond dissociation, which is consistent with the CO-insertion mechanism. On the other hand, at higher temperatures (180–220 °C) oxygenates were present in the products, which support the CO-insertion mechanism. So, it can be concluded that the CO-insertion and dissociation mechanisms may coexist and compete at normal FTS reaction temperatures. CO insertion mechanism is outlined in Figure 3.

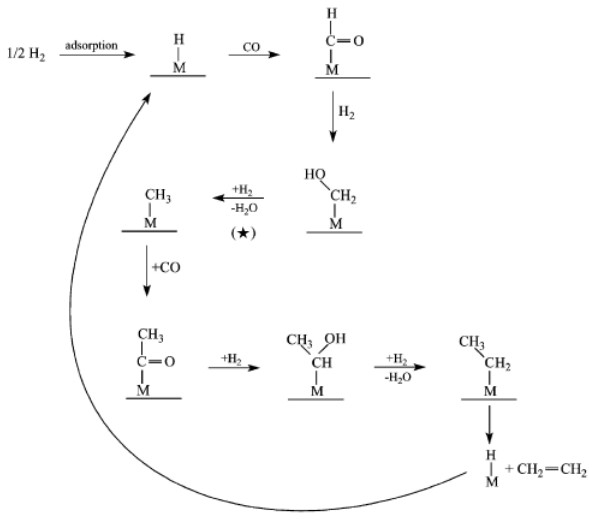

(★ rate controlling)

**Figure 3.** Schematic of CO insertion model. Reproduced with permission [13]. ★: rate controlling step.

Although the mechanisms of FTS have been discussed in detail in the past, during recent years, many steps have been taken to better explain this reaction. Quantum chemistry-based studies and isotopic kinetic experiments have played an important role in explaining FTS mechanisms. Recently, the H-assisted CO dissociation mechanism has gained tremendous attention for Fe and Co catalysts. To shed further light on this topic, more studies are needed to elaborate the electrostatic interaction between the feed molecules and the catalyst under reaction conditions.

## 3. Kinetics of FTS

The kinetic study of FT reaction is an important step in designing, optimization, and numerical simulation of industrial-scale processes. According to the open literature, various models and equations have been proposed for the explanation of the FT process by employing Fe and Co catalysts [8]. The major complexity in describing an accurate kinetic model to FT reaction is the various reaction mechanisms of FT and the large

number of species produced simultaneously. In fact, to achieve the ideal performance of FT process, a comprehensive kinetic study, which can explain product distribution, becomes necessary. Generally, the kinetic study of FTS can be done through two categories. In the first method, the focus is on the rate of syngas consumption and the product distribution is not considered because of the wide range of FT products. On the other hand, in second category, the distribution of the products is also considered. In both cases, the rate equations can be achieved empirically (e.g., power-law rate expression), semi-empirically or mechanistically (e.g., Langmuir-Hinshelwood-Hougen-Watson (LHHW) and Eley-Rideal (ER) rate equations). Moazami et al. [31] developed a comprehensive kinetics study on a $Co/SiO_2$ catalyst in FTS. They investigated the kinetics of the Co catalyst by empirical and mechanistic routes. The experimental data were fitted to the rate equations derived by power-law and LHHW models and the limitations and advantages of models were discussed. They found that the power-law model was able to explain the reaction kinetics only on the narrow range of operational conditions but the LHHW model was applicable on a wider range of operation conditions. The proposed kinetic models mostly depend on the type of catalyst, and operating conditions used in the FTS process and because of these factors, there is no unique kinetic model describing the consumption of syngas and products distribution [32]. The main difference between kinetic study of Co and Fe catalyst is the inactivity of the former in water–gas-shift (WGS) reaction. In LHHW and ER models, the detailed mechanism of FT kinetics can be achieved by considering appropriate sequential reaction pathways together with the assumptions about rate-determining steps (RDS). In most cases, the RDS is assumed to be the formation of the monomer [5]. The overall rate equation for each component ($H_2$, CO, $CO_2$, $H_2O$, and $C_nH_m$) consists of the sum of the reaction rates by FT and WGS reactions with the relevant stoichiometric coefficient. For example, $r_{co} = r_{FT} + r_{WGS}$ in which $r_{co}$ is the total rate of consumption of carbon monoxide and $r_{FT}$ and $r_{WGS}$ are the reaction rates of the FT and WGS reactions, which can be written as follows:

$$\text{(FT reaction) } nCO + (n + m/2)H_2 \rightarrow C_nH_m + nH_2O \tag{1}$$

$$\text{(WGS reaction) } CO + H_2O \rightleftharpoons CO_2 + H_2 \tag{2}$$

In Equation (1), n stands for the average carbon chain length of the hydrocarbon products and m is the average number of hydrogen atoms per hydrocarbon molecule. An important step in kinetics study is making sure that the internal and external mass transfers are not limiting the intrinsic reaction rate. To this end, the effects of interphase and intraparticle mass transport resistances should be examined using Weisz-Prater and Mears criteria for internal and external mass transfer limitations, respectively [33,34]. While the external diffusion rate can be increased by increasing gas superficial velocity, the internal mass transfer is improved by reducing the catalyst pellet size. An algorithm is outlined in Figure 4 for mass transfer and kinetics study of the FTS. Several reaction models have been proposed for Co and Fe catalysts in FTS; Table 3 summarizes some of the recent kinetics studies for FTS.

Various kinetic models have been developed for Fe- and Co-based FTS catalysts using empirical, semi-empirical and mechanistic approaches. To achieve an optimum kinetic model, nature of catalyst, reaction mechanism, operational conditions along with mass and heat transfer limitations should be concisely considered. Moreover, the different intermediate species should be taken into account for developing kinetics equations, which makes it challengeable in case of FTS. Mechanistic approaches based on LHHW models pave the way for improving the kinetics study of FTS catalysts.

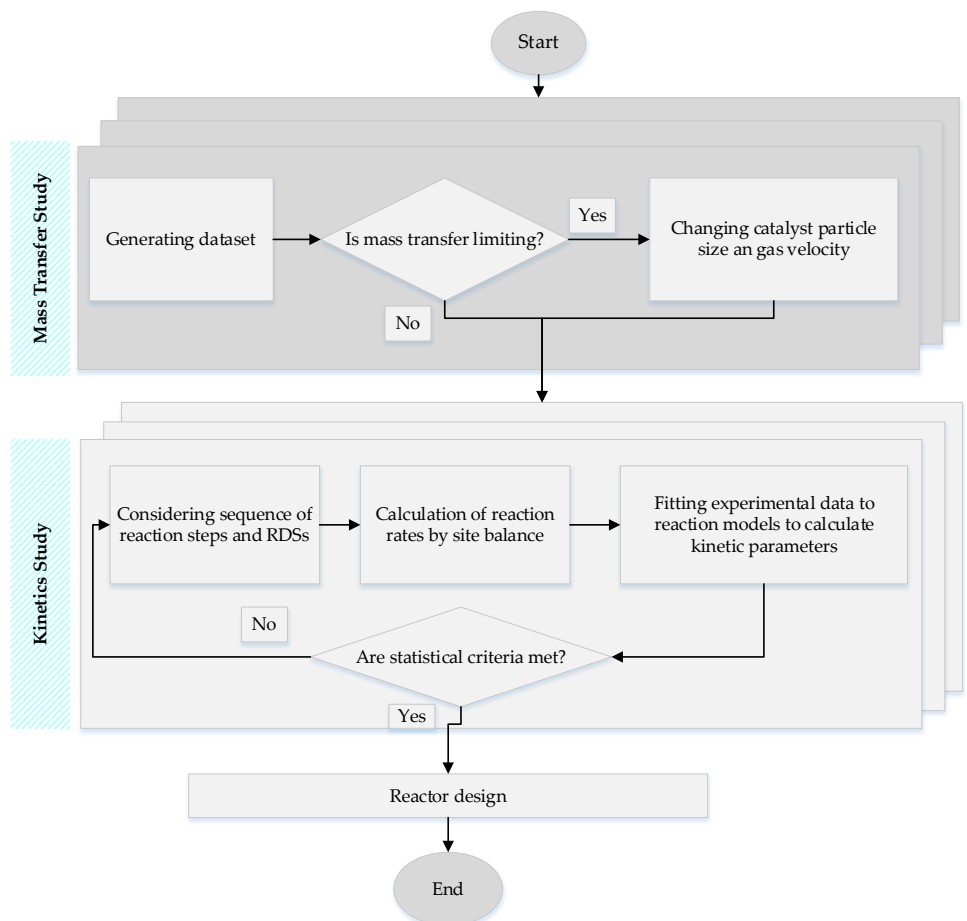

**Figure 4.** Flow chart of the kinetics study for FTS.

**Table 3.** FTS kinetic models based on iron and cobalt catalysts.

| Catalyst | Operational Conditions | Kinetic Model | Remarks | Reference |
|---|---|---|---|---|
| Co-Ce/SiO$_2$ | T = 200–300 °C, P = 0.1 MPa and H$_2$/CO = 1/1 and 3/2 | $-r_{CO} = \dfrac{kP_{H_2}}{(1+aP_{CO})^2}$ | - Based on LHHW approach<br>- E$_a$ = 31.57 kJmol$^{-1}$<br>- Kinetics study at constant pressure<br>- Limitations: It was assumed that CO is the predominant adsorbed species, and the surface coverage of other species were ignored. | [35] |
| Fe-Co/SiO$_2$ | T = 200–280 °C, P = 1–3 MPa and H$_2$/Co = 0.5–2.5 | $r_{FT} = \dfrac{kP_{CO}P_{H_2}^{\frac{1}{2}}}{\left(1+aP_{CO}+bP_{H_2}^{\frac{1}{2}}\right)^2}$ $r_{WGS} = \dfrac{k_W\left(P_{CO}P_{H_2O}-P_{CO_2}P_{H_2}/K\right)}{\left(1+aP_{CO}+cP_{H_2O}\right)^2}$ | - Based on LHHW and ER theories<br>- E$_a$ = 82.34 kJmol$^{-1}$<br>- Limitations: No information about the product distribution was reported. | [36] |
| Fe-Ni-Ce | T = 230–250 °C, P = 0.2–1 MPa and H$_2$/CO = 1 | $-r_{CO} = \dfrac{k_P b_{CO} P_{CO}(b_{H_2}P_{H_2})^2}{(1+2(P_{CO}b_{CO})^{0.5}+(b_{H_2}P_{H_2})^{0.5})^6}$ | - Based on LHHW and ER theories<br>- E$_a$ = 60.4 kJmol$^{-1}$<br>- Limitations: Kinetic study over the narrow range of temperature and water–gas-shift reaction was not taken into account in the developed model. | [37] |

**Table 3.** *Cont.*

| Catalyst | Operational Conditions | Kinetic Model | Remarks | Reference |
|---|---|---|---|---|
| Fe-Co-Ni | T=250–270 °C, P = 0.1–0.7 MPa and $H_2/CO$ = 1–2.5 | $-r_{CO} = \dfrac{k_P b_{CO} b_{H_2} P_{CO} P_{H_2}}{\left(1 + b_{CO} P_{CO} b_{H_2} P_{H_2}\right)^2}$ | - Based on LHHW approach<br>- $E_a$ = 79.88 kJmol$^{-1}$<br>- Significance of pore-diffusion limitations<br>- Limitations: Kinetic study over the narrow range of temperature and water–gas-shift reaction was not considered in the developed model. | [38] |
| Fe-HZSM5 | T = 300 °C, P = 1.7 MPa and $H_2/CO$ = 0.96 | $r_j = k_j P_{CO}{}^{m_j} P_{H_2}{}^{n_j}$ | - A 2D model of heat, mass, momentum, and kinetics was developed<br>- Determination of the optimum operating conditions and the tube specification<br>- The kinetic model was based on power-law and lumped reactions<br>- Limitations: Kinetics study at constant operating conditions. | [39] |
| K-Co/Al$_2$O$_3$ | T = 210–240 °C, P = 0.8 MPa and $H_2/CO$ = 1–3 | $-r_{CO} = \dfrac{k_2 K_1 P_{CO} P_{H_2}}{(1 + K_1 P_{CO})} -r_{CO} = k P_{CO}{}^{1.32} P_{H_2}{}^{1.42}$ | - Kinetics study by power-law and LHHW models<br>- $E_a$ = 138.5 kJmol$^{-1}$ (LHHW)<br>- $E_a$ = 87.39 kJmol$^{-1}$ (power-law)<br>- Limitations: Only investigated at constant pressure (0.8 MPa). It was assumed that CO was the predominant species occupied the total active site. | [40] |

## 4. Catalysts of FTS

FT catalysts are usually active for hydrogenation and metal carbonyl formation reaction under FT reaction conditions [41]. Among transition metals, only Fe, Co, Ni and Ru have these capabilities. Ru is an active catalyst for FTS, which produces long chain hydrocarbons without any promoters at relatively low reaction temperatures, but it is expensive and has limited availability. On the other hand, Ni has high selectivity to methane at higher temperatures, and forms volatile surface carbonyls at elevated pressures, which lead to deactivation of the catalyst [42,43]. Therefore, only Fe and Co can be considered suitable candidates for industrial FTS catalysts [44]. A brief comparison of the main characteristics of the four metals discussed above are shown in Table 4. Iron has a relatively low price, abundant availability, water-gas-shift (WGS) activity, which helps to counterbalance the lack or excess of hydrogen in the feed, selectivity to the lower olefins and flexibility to the process parameters such as temperature and pressure [45]. On the other hand, Co produces mainly linear alkanes and has higher activity as well as longer lifetime compared to Fe. In addition, Co has low WGS activity and is favored for natural gas-based syngas hydrogenation to high-molecular-weight hydrocarbons [5]. Another difference between these two catalysts is that Co catalysts are operated in only low temperature FT (LTFT at approximately 240 °C), which leads to formation of long chained hydrocarbons, while Fe-based catalysts can be active in severe operational conditions such as medium or high temperature conditions (MTFT and HTFT from 275 to 360 °C), which yields the production of lighter olefins and oxygenates [46]. While iron carbides are mainly responsible for activity and selectivity of Fe catalysts, cobalt carbides play an insignificant role in the activity of Co-based FTS catalysts. Moreover, cobalt carbide formation is inversely proportional to the $H_2/CO$ ratio and the reaction temperature [47]. For Co catalysts, the activity can be related

to the surface cobalt atoms measured by the amount of the adsorbed hydrogen or carbon monoxide on the reduced catalyst. Except for the iron catalyst, the same scenario does not work well, since a mixture of iron species such as iron carbides, oxides and metallic iron are active in FTS. Moreover, the WGS activity of iron catalyst, which competes with FT reaction, especially at higher CO conversion levels, leads to the different activity behavior compared to the Co catalyst. Another important difference between Co- and Fe-based FTS catalysts is the effect of promoters on the catalytic activity and selectivity. While it has been indicated that the selectivity of Fe catalyst can be highly affected by adding promoters, for selectivity of Co catalyst the promoters not only have limited effects but also some promoters have been reported to have detrimental effects on the catalyst activity [48].

**Table 4.** Overview of main characteristics of Ni-, Fe-, Co- and Ru-based catalysts [49].

| Active Metal | Price | FT Activity | WGS Activity | Hydrogenation Activity |
|---|---|---|---|---|
| Ni | expensive | low | low | very high |
| Co | expensive | high | low | high |
| Fe | cheap | low | very high | low |
| Ru | very expensive | very high | low | high |

As discussed, Fe and Co are the only suitable catalysts for industrial applications of FTS. These catalysts can be mainly differentiated by their different active phases during FTS and this affects the selectivity of catalysts. While the supported form of Co is preferred in industry, the precipitated form of Fe catalyst is widely used in gas-to-liquid (GTL) plants.

## 5. Support Materials for FTS Catalyst

The goal of support application in the FTS process can be summarized in three points: (1) providing a high surface area for dispersion of active site of the catalyst; (2) stabilization of the active phase under reaction conditions, and (3) providing suitable mechanical strength for the active phase of the catalyst as well as facilitating mass and heat transfer during reaction [5,50]. Therefore, selecting appropriate support is considered as one of the main steps during heterogeneously catalyzed reactions such as FTS. Moreover, because of the metal-support interaction, it is believed that optimum interaction affects the catalytic selectivity and activity. Strong interactions lead to the difficult reduction in the metal to its active phases, while weak interactions cause low dispersion of the active phase [44,51]. Metal oxides such as $Al_2O_3$, $SiO_2$, $TiO_2$, $ZrO_2$, and carbon materials are the most widely employed supports in FTS.

### 5.1. Alumina as a Support for FTS

Alumina ($Al_2O_3$) is one of the versatile supports employed in FTS due to its low price, high stability, and desired pore size distribution. Aluminas are generally synthesized by dehydration of aluminum hydroxides, and the $\gamma$-transition state of alumina ($\gamma$-alumina) exhibits the most optimal textural and chemical characteristics and hydrothermal stability for GTL (gas to liquid) applications. Textural properties and surface chemistry of alumina play an important role in FTS reactions. Ding et al. [52] realized that the presence of the acidic-basic hydroxyl groups in the structure of alumina-supported Fe-based FTS catalyst affected $C_{5+}$ hydrocarbon selectivity by improving particle size and dispersion of Fe catalyst. According to Xie et al. [53] the pore size of $Al_2O_3$ impacted the size of $Fe_2O_3$ particles; increasing pore size of the support led to the formation of larger $Fe_2O_3$ particles in the pores. They showed that the optimum pore size range for alumina was 7–10 nm, which corresponded to $Fe_2O_3$ particle sizes of 5–8 nm in the $CO_2$ hydrogenation reaction. Larger pore sizes reduce the number of active sites, and smaller pore sizes are unfavorable because of the difficult reducibility of iron particles. Surface modification of alumina is considered as an option for weakening metal–support interactions. Keyvanloo et al. [54] applied a novel $\gamma$-alumina doped with silica to study the effects of surface chemistry of support on activity and stability of the Fe catalyst in FTS. The pre-treatment of the synthesized

support at 1100 °C led to a lower number of acid sites and weaker metal oxide−support interactions, all desirable features for an effective FT catalyst. They observed that even after 700 h time on stream, the activity of the catalyst increased, and this interesting feature was due to specific preparation methods and the effects of silica to anchor the active sites of Fe to alumina, which suppressed sintering of catalyst. Modification of γ-alumina with magnesia (MgO) is another option that leads to the easier reduction of metal oxide and improved catalytic activity. It was shown that by the presence of a small amount of magnesia in γ-Al$_2$O$_3$, the activity of cobalt catalysts was improved in FTS. On the other hand, a decrease in the reduction of the catalysts was detected by using larger amounts of magnesia (>0.8 wt %), which was due to the formation of a hard reducible intermediate (MgO−CoO). The existence of this phase decreased the activity of the catalyst [55]. Although γ-alumina has high surface area, pore volume and acid/base characteristics, some undesirable features of this material such as dissolution of alumina during catalyst preparation (usually in aqueous media), limits its industrial application. These features are related to rehydration of γ-Al$_2$O$_3$ during catalyst implementation, with the H$_2$O produced in the FT process and thermal degradation due to the sintering and phase transformation in the catalyst regeneration step due to hotspots [56]. Strong metal–support interactions lead to the lighter range of hydrocarbons in FTS. Snel (1989) [57] reported that in the case of an alumina-supported iron catalyst, the products were significantly lighter than the products from the unsupported iron catalyst.

### 5.2. Silica-Supported Catalyst for FTS

Silica gel, colloidal silica, kieselguhr, and fumed (pyrogenic) silica are four types of silica materials [58]. In comparison to alumina, silica has weaker interactions with the active metal, which facilitates metal reduction [59]. Periodic mesoporous silicas are other types of silicate materials which are used as support of FTS. Due to narrow pore size distributions, high surface areas, pore volumes, and controllable acid/base properties, periodic mesoporous silicas can improve the textural properties of the support, thus impacting the productivity and hydrocarbon selectivity of the catalyst in FTS [60]. Mobil Catalytic Material number 48 (MCM-48, cubic 3D porous structure), Mobil Catalytic Material number 41 (MCM-41), SBA-15, hexagonal mesoporous silica (HMS) and silica hollow sphere (SHS) are from this group of silicas. Application of these supports leads to better understanding about the effect of the textural properties of the support on the catalyst behavior. Cano et al. showed that iron catalyst supported on mesoporous solid SBA-15 led to the small iron oxide particles with a narrow size distribution, which improved catalyst activity and selectivity in FTS [61]. In another work, Cano et al. studied the importance of porosity in silica supports for catalysts of FT reaction. According to the catalytic tests, Fe/SBA-15 showed a higher activity, major chain-growth formation of the products and more selectivity to olefins than the conventional Fe/SiO$_2$ catalyst [62]. Molecular sieves such as SBA-15 due to their mesoporous structure, are expected to lower the diffusion limitation that a microporous material typically experiences, and this affects the reaction rate in FTS. Cano et al. [63] analyzed the catalytic behavior of two support materials for iron, the first support was SBA-15, and the second one was MCM-41. The higher activity of the SBA-15 supported catalyst with larger iron particles was attributed to the "structural sensitivity" of FT synthesis at nanoscale range, this means that the activity of the solid catalyst is a function of the crystal size of the active phase, generally in the 1–10 nm range. Considering the advantages of these mesoporous materials there are some limitations in employing them as support for the FTS catalyst. For instance, preparation of MCM-41-supported cobalt catalyst by impregnation and drying led to the loss of the long-range order in the hexagonal mesoporous structure, which negatively affected the surface area and pore volume of the final catalyst [64].



### 5.3. Carbon-Based Supports for FTS

Carbon-based supports due to higher surface areas, suitable porous structure and stability at high-temperature reaction conditions have drawn interest in FTS. It has been suggested that carbon-based supports, due to their inertness, can circumvent the deactivation problem of metal oxide-supported catalysts owing to strong metal–support interaction (SMSI) [5]. The inert surface of the carbon materials facilitates the reduction of the metal precursor to its active phases. Furthermore, the effects of metal particle size in FTS can be investigated using carbon supports because of their weaker interaction with the active metal [65,66]. Various types of carbon materials consist of graphite, carbon black, activated carbon (AC), carbon nanotube (CNT), carbon fibers, ordered mesoporous carbon (OMC), and carbon encapsulated metal nanoparticles (CEMNPs) that have been employed as support of Fe- and Co-based FTS catalysts. Functional groups present in the structure of the carbon materials such as oxygen-containing groups can improve the metal-support interaction leading to improved selectivity of the catalyst to desired products in FTS [67]. Activated carbons are the most widely employed carbon supports in heterogeneously catalyzed reactions. Activated carbon can be produced from coal, wood, polymer, and agriculture residues by chemical or physical activation route [58,68]. One of the most exciting properties of the carbon supports such as activated carbon, is its electronic conductivity because of the delocalized $\pi$ electronics. This leads to a close contact between the metal microcrystals and carbon matrix because of electronic interactions. Considering all advantages of using AC as support material of the FTS catalyst, it has been shown that mesoporous materials with pore size of 10–15 nm could provide suitable texture for the formation of metal crystallites and this criterion limits the efficient use of activated carbon in FTS because of its microporous structure [44,69]. These limits led to the introduction of other types of carbon materials such as carbon nanotubes (CNTs) and nanofibers (CNFs), which have merits of reproducibility, mesoporosity and well-defined structures. Abbaslou et al. [70] evaluated the effects of pore diameter and structure of the iron catalysts supported on carbon nanotubes (CNTs) on the activity and selectivity of the catalyst. They found that iron oxide particles on the catalyst supported on CNT with wide pore structure (17 nm) were larger than those on narrow pore samples (11 nm). In terms of activity of the catalyst, for the catalyst with narrow pore CNT, the activity was 2.5 times that of the wide pore CNT-supported catalyst, and the wide pore CNT-supported catalyst was more selective towards lighter hydrocarbons. Carbon-encapsulated metal nanoparticles (CEMNs) are another group of carbon material, which consists of metal particles in the core and a multilayer of carbon shells around the core. These carbon layers have an average interplanar distance of 0.34 nm. High mechanical, thermal, and chemical stabilities of CEMNs have made them promising candidates for FTS [71,72]. Moreover, owing to the electronic interaction between the metal particles inside the core and the graphitic carbon shells, application of CEMNs is speculated to increase the catalytic activity and stability [73].

### 5.4. Other Support Materials for FTS

Titania is another metal oxide carrier which has been employed in FTS. Titania naturally appears in three crystalline forms: anatase, rutile and brookite, the former two are of considerable interest owing to the larger surface area of the anatase phase and the greater mechanical strength of the rutile phase. Rutile is a more thermally stable form of titania [58]. $TiO_2$ is considered as one of the suitable supports for FT-based catalysts due to its low cost, safety, and chemical stability. Abrokwah et al. [74] carried out a study to analyze the effect of $TiO_2$ support on catalytic performance of Fe, Co, and Ru metals in FTS. According to the results of Temperature Program Reduction (TPR) analysis, Fe- and Co-supported titania catalysts showed strong metal–support interaction (SMSI) compared to $Ru/TiO_2$, which indicated weaker metal–support interaction. On the other hand, the rutile phase was absent in both Fe- and Co-supported catalysts, which led to lower stability of these catalysts compared to $Ru/TiO_2$ catalyst due to the presence of a rutile phase in its structure. MgO is another support considered for FTS-based catalysts. Gallegos et al. [75] have reported that

Fe/SiO$_2$−MgO catalyst increases the rate of hydrocarbon formation and they also indicated that the addition of MgO increased the selectivity to olefins and suppressed the selectivity of methane. Zirconium oxide is another carrier, which has been shown to increase the selectivity towards C$_{5+}$ hydrocarbon production. Moreover, zirconium oxide acts as a promoter for supported catalysts in FTS and improves the catalyst activity [76]. Another distinctive characteristic of zirconia is the absence of Brønsted acid sites, and all the acidity is related to Lewis sites, which are active for hydrogenation reactions such as FTS. Van den Berg et al. [77] studied the reduction behavior of Fe/ZrO$_2$ and potassium-promoted Fe/ZrO$_2$ catalysts in FTS. They detected high initial dispersions in both catalysts. The iron particles in the potassium-promoted catalyst reduced more easily than unpromoted catalyst. As a result of divalent iron formation and reaction of this specie with zirconia, a stable mixed oxide was formed. This mixed oxide is suggested to maintain a high dispersion of the metallic iron particles. CeO$_2$ is another metal oxide that can act either as a promoter or stabilizer of the alumina-supported catalyst to improve its thermal stability. Moreover, CeO$_2$ has the capability of scavenging deposited carbons and this improves the stability of the catalyst. Like other metal oxides, ceria (CeO$_2$) is prone to sintering and catalyst deactivation especially at high temperature. It has been reported that modification of the support materials such as alumina with La$_2$O$_3$ and CeO$_2$ could alter the surface properties of the catalyst to achieve suitable metal–support interactions [76,78]. Recently, Munirathinam et al. [79] introduced hydroxyapatite (HAP, (Ca$_{10}$(PO$_4$)$_6$(OH)$_2$)) as a new support material for FTS over Co catalyst. According to their investigation Co/HAP showed higher activity and selectivity to C$_{5+}$ hydrocarbons (82–88%) compared to Co/Al$_2$O$_3$ catalyst in FTS at 20 bar, 220 °C, and H$_2$/CO = 2. High activity and selectivity to C$_{5+}$ are due to the high thermal, mechanical strength, low water solubility, tunable porosity, and acid-basic features of HAP.

The effective design of support materials for FTS catalysts plays a leading role in increasing the catalyst activity and selectivity towards desired products. Controlling the metal–support interaction is one of the main factors affecting optimum catalyst design. Although it is believed that supports are not involved in FT reaction, some supports such as zeolites, further discussed in Section 9, are crucial for cracking and isomerization reactions. Recently, research attention has been drawn to new materials such as HAP as a support of FTS catalyst due to their optimum physicochemical properties.

## 6. Catalyst Preparation

Sol–gel, precipitation, impregnation, ion-exchange, carbon-vapor deposition, spray-drying, and plasma-spray technologies are among the most employed preparation techniques of FTS catalyst [80–83]. The synthesis method affects the morphology, catalyst particle size and surface area of the final catalyst to a large extent. Tasfy et al. [84] indicated that, compared to precipitation method, the average particle size of the Fe particles supported on SiO$_2$ was smaller when the catalyst was prepared by the impregnation. In another work, Sarkari et al. [85] indicated the effects of the preparation method on the catalytic activity of bimetallic Fe-Ni catalyst supported on alumina. According to their work, the impregnated catalyst had higher activity and selectivity towards light olefins and C$_{5+}$ products than the co-precipitated catalyst. Alayat et al. [86] prepared a nanostructured iron catalyst supported on silica nano spring through sol–gel, precipitation, and impregnation techniques. The highest CO conversion (76%) of FTS was for the catalyst that was prepared by impregnation and activated under CO stream before FT reaction. Plasma-spray technology is another catalyst preparation method that has gained attention. Traditional catalyst preparation methods are labor-intensive and involve several stages which is time-consuming. On the other hand, the plasma technique consists of a single step with a low number of variables to be controlled. The uniformity in quality of materials, achieving highly distributed and, consequently, smaller nanometric size metal particles, as well as enhanced catalyst lifetime are advantages in applying plasma technology as a catalyst preparation method [87–89]. For example, Aluha et al. [88] synthesized carbon-

supported Fe and Co catalysts by a single-step plasma technique for FTS. According to their results, Fe/C and Co/C catalysts showed 30 and 20% CO conversion, respectively, higher than that of catalysts which were prepared by impregnation and precipitation techniques under similar reaction conditions.

Catalyst preparation is one of the fundamental steps in designing an active catalyst for FTS. Impregnation and precipitation are two techniques mostly used for the FTS catalyst preparation step. Preparation method can significantly affect the particle size and surface area of the final synthesized catalyst, which determines the FTS activity and selectivity. The use of plasma and microwave radiation has drawn much attention during last several years for preparation of FTS-based catalysts due to their effects on enhancing the catalyst activity, stability, and selectivity.

## 7. Catalyst Characterization

Characterization is an important step to understand the nature of the active catalyst and designing effective catalysts. It can be carried out by spectroscopy, microscopy, or diffraction methods. All mentioned techniques are based on studying the changes in sample after being subjected by an incident beam, which can be composed of photons, electrons, ions, neutrals, or magnetic, electric, acoustic, or thermal fields. Another group of characterizations are thermal analyses that are employed to scrutinize the heat involved in reactions and evaluate thermodynamical properties of catalysts [90]. In situ and operando techniques employing synchrotron radiation for the characterization of FTS-based catalysts play a pivotal role in understanding the chemical properties of the catalysts and their relationship with the FTS reactions. Using both hard and soft X-rays in synchrotron-based methods has gained tremendous attention over the past two decades for shedding light onto the mechanism and structure of FTS catalysts. Catalyst activation, effects of the promoters and catalyst stability are among the important areas that have been focused on through characterization techniques using synchrotron radiation. Coupling a standard temperature-programmed-reduction (TPR) method with X-ray near absorption edge structure (XANES) and extended X-ray absorption fine structure (EXAFS) spectroscopies surmount the ambiguities of the reduction process of Fe and Co catalysts. Jacobs et al. [91] have employed TPR-EXAFS/XANES techniques to scrutinize the nature of chemical transformations during activation of alumina- and silica-supported Co catalysts in H2 atmosphere. Unlike standard TPR, which makes some assumptions about the Co species during reduction in $H_2$ atmosphere, the synchrotron-based technique directly provided information not only on the nature of Co species but also clarified the relationship between the crystallite size and degree of metal–support interactions. According to their results, higher reduction temperatures were needed for reduction of cobalt oxide particles supported on alumina in comparison with unsupported $Co_3O_4$ or only weakly interacting supported cobalt catalyst (silica). The effects of promoters using XANES and EXAFS spectroscopies have been investigated using synchrotron techniques. These techniques enable a better understanding related to the effects of promoters on carburization rate of iron and reduction behavior of Co catalysts [92,93]. Thermal stability of Fe- and Co-based catalysts has also been investigated by synchrotron-based characterization methods. These techniques provide information about the mechanism of deactivation for Co catalyst and the temperature effects on the stabilization of iron carbide phase against oxidation [91]. Herein, some of the most common spectroscopy, diffraction and microscopy-based characterization techniques are discussed for Co and Fe catalysts used in FTS.

### 7.1. Diffraction-Based Characterizations

Diffraction methods are mainly employed to study the crystallographic structure of the catalyst. X-ray diffraction (XRD) is one of the most applied methods for studying the crystalline structure of the Fe and Co catalysts in FTS. X-ray diffraction occurs because of elastic scattering of X-ray photons by atoms in a periodic lattice. XRD can also be carried out in situ because of the high penetrating power of X-rays. One of the main

limitations of XRD is that clear diffraction peaks are only present when there are sufficient long-range order particles in the sample [94]. Table 5 summarizes some of the objectives of characterizing Fe and Co catalysts in FTS by using XRD analysis.

**Table 5.** Characterization of Fe- and Co-based FT catalysts by XRD technique.

| Catalyst | Focus | Reference |
|---|---|---|
| Fe/CNT | Determination of Fe particle size doped inside and outside of the carbon nanotube (CNT) | [95] |
| K-Fe/graphite | Determining phase evolution of Fe, effects of K on carburization of Fe and formation of high molecular weights hydrocarbons | [96] |
| Fe-SiO$_2$ | Relation between Fe$_2$O$_3$ particle size and pore diameter of silica | [97] |
| Co/TS-TiO$_2$ | Investigating the chemical composition of the catalyst, Co$_3$O$_4$ crystallite size and different phases of TiO$_2$ | [98] |
| Co-Fe/TiO$_2$ | Determination of the weight fraction of rutile in the support, good dispersion of the metal oxides, strong interaction between support and Fe-Co, alloy formation and particle size | [99] |
| Y-Co/mAl$_2$O$_3$ | Effect of promoter on crystallite size of Co oxide, relation between crystallite size and Co-support interaction | [100] |

### 7.2. Spectroscopy-Based Techniques

Spectroscopy deals with the interaction between matter and electromagnetic radiation as a function of the wavelength or frequency of the radiation. Vibrational and X-ray-based spectroscopies are among the most employed characterizations for Co and Fe catalysts in FTS. A vibrational spectroscopy study shows that the transitions occurs between energy states of the molecule because of placing in an electromagnetic field [101]. Raman and infrared (IR) spectroscopies are two of the most common vibrational-based techniques for analyzing FTS-based catalysts. On the other hand, X-ray-based spectroscopies analyze the chemical and electronical structure of the catalyst. X-ray photoelectron (XPS) and X-ray absorption (XAS) are among the most suitable X-ray-based methods [102]. Another robust spectroscopy technique is called Mössbauer spectroscopy, which studies the absorption of a $\gamma$-photon by a nucleus in ground state. Table 6 shows the summary of the some of the spectroscopy techniques for characterizing Fe and Co catalysts in FTS.

**Table 6.** Spectroscopy-based analysis for Fe and Co catalyst in FTS.

| Catalyst | Technique | Focus | Reference |
|---|---|---|---|
| K-Mn-Fe/SiO$_2$ | IR | - Investigating the relation between surface adsorbed species and reduced iron phases | [103] |
| Ru-Co/TiO$_2$ | In Situ FTIR | - Explanation of CO adsorption on different sites of Co<br>- Studying the evolution of the surface Co species during FTS<br>- Effect of metal–support interaction in unpromoted catalyst on blockage of surface Co species | [104] |
| CEINPs | Raman spectroscopy | - Degree of graphitization<br>- Relation between uniform carbonaceous structure and thermal treatment | [105] |
| Co/GNS | Raman spectroscopy | - Investigating the ratio of disordered to graphitic-like carbon structure<br>- Increase in the defected sites by functionalization of the support | [106] |

**Table 6.** *Cont.*

| Catalyst | Technique | Focus | Reference |
|---|---|---|---|
| Mn-K-Cu-Fe/mAl$_2$O$_3$ | XPS | - Revealing oxidation states of the catalyst<br>- Calculation of atomic percentages of iron and promoters on the surface of the catalyst | [107] |
| Fe/NS | XPS | - Evaluation of phase composition<br>- Distinguishing between $\gamma$-Fe$_2$O$_3$ and $\alpha$-Fe$_2$O$_3$ phases by means of satellite peaks | [86] |
| Ru-Co/Al$_2$O$_3$-SiO$_2$ | XPS | - Detection of chemical states of cobalt and ruthenium in mixed supported catalyst<br>- Indicating interaction of Co metal with different types of support by analysis of Co 2p$_{3/2}$ and Co 2p$_{1/2}$ peak intensities<br>- Studying the effect of metal–support interactions on degree of exposure of the active sites | [108] |
| Mn-K-Cu-Fe/mAl$_2$O$_3$ | XAS | - Presence of electronical interaction between Fe and promoters by using X-ray near absorption edge structure( XANES) analysis and the effect of this interaction on FT activity<br>- Evaluation of the influence of promoters on reduction of Fe | [107] |
| Co/SiO$_2$ | XAS | - Identifying the coordination of Co atoms<br>- Studying the degree of reduction of Co as well as three different phases of Co (metallic Co, CoO and Co$_2$SiO$_4$) and their composition | [109] |
| Ru-Co/TiO$_2$ | In situ XAS | - Scrutinizing the change in local environment of Ru particles during FTS reaction because of interaction with adsorbed species<br>- Exploring the coordination numbers | [104] |
| Fe/CNF | Mössbauer spectroscopy | - Identifying the relation between the active phase and catalyst activity, difference between carbided Fe in promoted and unpromoted catalyst<br>- Exploring a correlation between percentage of Fe carbide species and catalyst activity | [110] |
| Na-Mn-Fe (microsphere) | Mössbauer spectroscopy | - Phase identification of Fe by Mössbauer parameters, promotion effect of Na-Mn on transformation of magnetite to $\chi$-Fe$_5$C$_2$<br>- Analyzing the effect of Mn on selectivity of catalyst towards light olefins formation | [111] |

*7.3. Microscopy Based Characterizations*

Microscopy is one of the main categories of heterogeneous catalyst characterization, which maps the surface and sub-surface of the material. Microscopy techniques use photons, electrons, ions, or physical cantilever probes to examine the structure of sample. Scanning electron microscopy (SEM) and transmission electron microscopy (TEM) are among useful microscopy techniques for probing catalyst particle size, dispersion, elemental composition, and morphology [112]. Table 7 outlines some of the recent achievements in microscopy-based techniques for characterizing Fe and Co catalysts in FTS.

**Table 7.** Microscopy-based techniques for analysis of physicochemical properties of Fe- and Co-based FT catalysts.

| Catalyst | Technique | Focus | Reference |
|---|---|---|---|
| Co/CNT | TEM | - Revealing the presence of carbon shells around metal nanoparticles with different thicknesses<br>- Determination of Co particle size distribution<br>- Studying the effect of sintering temperature on particle size of Co particles<br>- Inability of TEM in detecting the structure of ultra-small particles | [113] |
| Fe@C (MOF) | SEM | - Analyzing the morphology of samples before and after pyrolysis at 700 °C and reduction under different atmospheres which showed no significant difference | [114] |
| Co/Char | TEM | - Presence of bimodal size distribution with different morphologies for Co particles | [115] |
| Fe/SBA-15 | SEM and TEM | - Studying the textural properties of catalyst<br>- Detection of no significant difference between shape of particles in support itself and the catalyst according to SEM<br>- Presence of most iron particles inside the pores of support<br>- Revealing the hexagonal pore structure characteristic of the SBA-15 with non-uniform pore size distribution ( PSD) according to TEM | [116] |
| Co/SiO$_2$ | SEM and TEM | - Detection of better porosity development in the catalysts prepared by microwave-assisted technique compared to conventional methods by SEM<br>- Existence of ideal particle size (10 nm) for FTS by microwave-assisted prepared catalyst compared to narrow particle sizes (2–3 nm) of conventionally dried catalysts by using TEM | [117] |

### 7.4. Thermal Methods

Thermogravimetric analysis (TGA), temperature-programmed desorption (TPD) and temperature-programmed reduction (TPR) are useful techniques for investigating the chemical structure of Fe and Co catalysts in FTS. These characterizations analyze the mass variation, reduction behavior and active sites on the catalyst surfaces and help to understand the mechanisms of catalytic reactions including adsorption, surface reaction and desorption [118]. Table 8 illustrates some of the works done by employing mentioned techniques to characterize Fe and Co catalysts in FTS.

**Table 8.** Characterization techniques based on thermal methods for Fe and Co catalyst in FTS.

| Catalyst | Technique | Focus | Reference |
|---|---|---|---|
| Mn-K-Fe/SiO$_2$ | H$_2$-TPD | - Analyzing the chemisorption behavior of iron species according to desorption peaks of H$_2$ from different active sites of iron | [103] |
| Ru-Co/SiC-Al$_2$O$_3$ | H$_2$-TPR | - Investigating the reduction of Co species and Co$_x$O$_y$-Al$_2$O$_3$<br>- Estimation of the degree of reduction (DOR)<br>- Analyzing the effect of SiC on weakening the support–Co interactions | [119] |
| Co/HAP and Co/Al$_2$O$_3$ | H$_2$-TPR | - Identifying two-step reduction behavior for the catalyst<br>- Decrease in reduction temperature of Co/HAP compared to Co/ Al$_2$O$_3$ due to the absence of small cobalt oxide particles and refractory Co-aluminate species which are hard to reduce | [79] |
| Ba-Co/Al$_2$O$_3$ | CO-TPD | - Detection of decrease in Co dispersion in Ba-modified catalysts because of increase in sintering of Co particles<br>- Covering of the surface of the catalyst by Ba which diminished fraction of real exposed Co surface and<br>- Improving effect of Ba on facilitating the adsorption and dissociation of CO | [120] |

**Table 8.** *Cont.*

| Catalyst | Technique | Focus | Reference |
|----------|-----------|-------|-----------|
| N-doped Co/HCSs | TGA | - Investigating thermal stability of the catalysts in FT reaction conditions for N-doped and N-free catalysts | [121] |
| Fe-Co/TiO$_2$ | TGA | - Investigating the retention of hydrocarbon products on the used catalyst<br>- Detection of weight increase due to the oxidation of reduced species<br>- No weight loss for monometallic catalysts<br>- 8–10% weight loss for bimetallic catalysts<br>- Weight loss because of combustion of carbon deposits | [99] |

Particle size, metal dispersion and free metal specific surface area are among the important parameters affecting the activity and selectivity of Fe and Co catalysts during FTS, which can be determined by H$_2$ and CO chemisorption techniques. Although the particle size of the catalyst can be evaluated by XRD and TEM, the presence of amorphous phases in case of XRD and tedious procedure of TEM limit their application. Traditional ex situ H$_2$-chemisorption on the other hand, accounts for the sites that are only active for H$_2$ adsorption, i.e., a catalyst with low degree of reduction leads to the low number of active sites according to H$_2$-chemisorption. However, when increasing the time-on-stream in the FTS process, more active sites may be accessed, possibly increasing CO activity and product selectivity. Yang et al. [122] have investigated the effect of Co particle size on activity and selectivity of FTS by in situ CO chemisorption by different supports. They found that Co particle size based on an in situ CO chemisorption technique was in good agreement with turnover frequencies (TOFs) under reaction conditions. They also detected that the support plays an important role for modifying the properties of Co particles. Like cobalt, the activity and selectivity of iron catalyst depend on the particle size. Park et al. [123] studied the effect of particle size of iron oxide on catalytic activity and selectivity of alumina-supported Fe catalyst in FTS. They employed XRD, TEM and CO chemisorption techniques for evaluation of particle size of the iron catalyst. The particle size results achieved by three techniques were in good agreement with each other. Moreover, they detected that by increasing particle size from 2.0 to 6.1 nm the TOF increased from 0.06 to 0.187 s$^{-1}$.

## 8. Selectivity of Products towards Liquid Fuels in FTS

Compared to other fuels such as dimethyl ether and hydrogen, liquid fuels have higher energy. Therefore, synthesis of these fuels by Fischer–Tropsch process, which provides a route for formation of these fuels with a small amount of sulfur and aromatics, plays a pivotal role in the economization of this process. The liquid fuels that can be achieved through FTS, are gasoline C$_{5-11}$, jet fuel C$_{8-16}$ and diesel fuel C$_{10-20}$. FTS yields a mixture of hydrocarbons, which needs to be further upgraded to fuels and chemicals as final products. Hydrocracking is considered as a post-treatment process for FT products in GTL plants, which cracks heavy FTS products such as wax in the presence of hydrogen and increases the selectivity of desired products such as diesel and jet fuel blends. Hydrodimerization is also another post-treatment process producing synthetic lubricant base oils [124]. The schematic of the shell GTL plant is outlined in Figure 5, which contains a hydrocracking unit. Direct synthesis of liquid fuels can be achieved by tuning the product selectivity of FT catalyst and it is one of the most challenging parts of this process due to the various reaction mechanisms and wide range of products formed during reaction [101,125]. Products distribution of FTS can be predicted by a statistical method known as Anderson–Schulz–Flory (ASF) distribution according to the equation below [126]:

$$\log \frac{W_n}{n} = n \, \log \alpha + \log \frac{(1-\alpha)^2}{\alpha} \qquad (3)$$

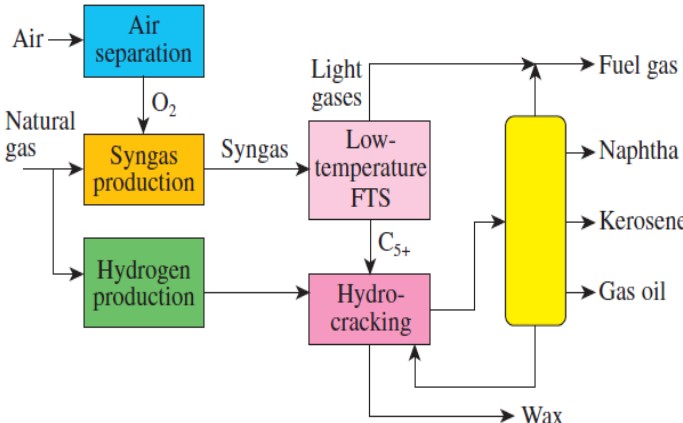

**Figure 5.** Schematic of CO simplified schematic of Shell gas-to-liquid (GTL) plant. Reproduced with permission [124].

In Equation (3), $W_n$ represents mass fraction of a particular product, n is the carbon number and $\alpha$ is the probability of chain growth. Although ASF equation can predict the formation of hydrocarbons in FTS, there are some deviations of light hydrocarbons including methane from this statistical distribution. For instance, it fails in effective analysis of the bifunctional catalytic systems that can also catalyze the C–C cleavage of heavier hydrocarbons [127]. Moreover, methane is usually produced in excess amounts as compared to the predicted value by ASF, especially for Co and Ru catalysts due to the hydrogenolytic cleavage of the 1-olefins in secondary reactions. On the other hand, $C_2$ and $C_3$ yields are is often lower than the predicted value for Fe- and Co-based catalysts. The variation in $C_2$–$C_4$ distribution is assumed to be related to the incorporation of the $C_2$–$C_4$ olefins into the chain [128]. Therefore, there is limitation in predicting the maximum attainable selectivity of gaseous and liquid fuels by ASF distribution, which is about 30–45%. Thus, ASF distribution seems to be non-selective for a desired range of hydrocarbons [129,130]. For predicting the products distribution of bifunctional catalysts, the only chain growth factor ($\alpha$) in traditional ASF, cannot effectively explain the wide range of products. Another parameter, known as r and called cracking degree ($\beta$), has been introduced to account for the cracking ability of the bifunctional catalysts. Li et al. [131] have investigated the new models for distribution of gasoline, jet fuel, and diesel fuel by using a mesoporous Y-type zeolite-supported Co catalyst in FTS. Based on this approach, the selectivity of different fractions was successfully predicted by chain growth ($\alpha$) and cracking ($\beta$) factors. The cracking degree ($\beta$) for gasoline, jet fuel and diesel fuel were 0.7, 0.8 and 0.6, respectively. Moreover, the experimental selectivity of three mentioned fractions of products were in good agreement (within 5% error) with the results of bifunctional catalyst distribution model. Generally, the nature of active metal, support, promoters, reactor design and the operating conditions affect the selectivity of products in FTS, which will be discussed in this review.

### 8.1. Effects of the Nature of Active Components, Support, and the Promoters

Regarding the active phases of Fe and Co metals, it has been shown that iron carbide and metallic Co are the active phases for chain growth during FT reaction [129]. While the metallic cobalt species are responsible for increasing the product selectivity of FT reaction towards $C_{5+}$, the strategy of iron carbide species is not clear for the iron catalyst. In fact, during FT reaction for the iron-based catalyst, a mixture of phases including $Fe_3O_4$, metallic Fe, and Fe carbides exist, which is the result of reconstruction of the catalyst under syngas atmosphere [132]. Another key parameter for controlling the selectivity of FT products is the nature of the support. Metal–support interaction and physicochemical properties of the support significantly affect the product selectivity of FTS [129]. The promoter is another factor, which influences the selectivity of FT products. These materials are added to the

catalyst in very small amounts (~1–2%wt) to improve its catalytical, structural, electronical, and textural characteristics or can act as stabilizers and poison-resisting agents. Generally, promoters are classified as textural or structural groups. While the former acts as a physical effect, the latter has more chemical effects on the catalyst [58]. Table 9 summarizes some of the research carried out to increase the selectivity of Fe- and Co-based catalysts in FTS towards $C_{5+}$ hydrocarbons.

**Table 9.** Effects of active metal, support, and promoter nature on $C_{5+}$ selectivity of Fe- and Co-based FT catalysts.

| Catalyst | Operational Conditions | Focus | Reference |
|---|---|---|---|
| $Fe_2O_3$@$MnO_2$ | T = 280 °C, P = 2 MPa and $H_2$/CO = 1 | - $C_{5+}$ selectivity of the catalyst increased from 44.6 to 66.6 wt% for Mn promoted catalyst.<br>- Mn facilitated CO dissociation and chain growth.<br>- Methane selectivity decreased from 16.8 to 8.9 wt% by Mn promotion. | [133] |
| $Al_2O_3$-Co/$SiO_2$ | T = 214 °C, P = 2 MPa and GHSV = 1000 h$^{-1}$ | - Promoting the catalyst with 1 wt% of alumina increased CO chemisorption.<br>- $C_{5+}$ selectivity of the alumina-doped catalyst increased from 77.4 to 80.1 wt%. | [134] |
| Mesoporous Fe spindles | T = 280 °C, P = 2 MPa and $H_2$/CO = 1 | - The effect of pore size of the unsupported catalyst on FT activity and selectivity was investigated.<br>- Employing the active phase assembled mesoporous structure to tune the selectivity of the catalyst to $C_{5+}$ formation<br>- $C_{5+}$ selectivity reached 65wt%.<br>- Larger pores and the nanospaces developed in the structure of the catalyst, positively affected selectivity, and diffusional limitations. | [135] |
| Co/$Al_2O_3$ and Co/SiC | T = 220 °C, P = 4 MPa and $H_2$/CO = 2 | - Higher $C_{5+}$ selectivity of silicon carbide-supported catalyst compared to alumina-supported one (80 wt% vs. 54 wt%)<br>- Improved $C_{5+}$ selectivity of Co/SiC was due to the high heat removal efficiency of SiC compared to alumina.<br>- Positive effect of the coexisting meso and macro-pores in SiC on $C_{5+}$ selectivity | [136] |
| Fe/CNT-$MnO_x$ | T = 270 °C, P = 2 MPa and $H_2$/CO = 1 | - CNT-$MnO_x$ nanocomposite led to high $C_{5+}$ selectivity (up to 93.8%) due to the distinctive geometric structure of support, moderate metal–support interaction, and Mn promotion effect.<br>- High WGS activity of Mn promoted catalyst | [137] |
| Co/Al-SBA-15 | T = 230 °C, P = 1 MPa and $H_2$/CO = 2 | - Introduction of the acid sites with proper strength, to the SBA-15 supported catalyst improved the selectivity for $C_8-C_{18}$ products from 43.9 to 52.4%.<br>- Addition of Al decreased the selectivity of heavy products because of Brönsted acid sites<br>- By increasing the acidity of the catalyst, selectivity shifted towards light products. | [138] |

*8.2. Effects of Process Conditions*

Process conditions play a pivotal role in tunning the selectivity of FT products. Temperature is one of the primary process parameters that can control the selectivity of FT products. It has been indicated that by an increase in temperature the products of FTS shift towards lower carbon number species, for either cobalt-based or iron-based FTS catalysts [5,139,140]. Niu et al. [141] investigated the effects of process conditions on selectivity of an industrial cobalt-based catalyst in FTS. They detected that lower temperature and higher pressure led to decrease in methane selectivity and increase in $C_{5+}$ selectivity.

Partial pressure of CO and $H_2$ is another variable that can affect selectivity of products. According to the open literature, product selectivity changes to heavier products and to more oxygenates with increasing total pressure. On the other hand, by increasing $H_2$/CO ratio, lighter hydrocarbons are favored [139,140,142]. Savost'yanov et al. [143] reported that increased total pressure (up to 6 MPa) had a positive effect both on CO conversion and $C_{5+}$ selectivity of Co-$Al_2O_3$/$SiO_2$ catalyst. They also indicated that at high pressure (6 MPa) the catalyst was more prone to deactivation compared to operation at more moderate pressure of 2 MPa. Todic et al. [144] analyzed the effects of operational conditions on the performance of the Fe-Cu-K/$SiO_2$ catalyst for FTS. They found that increasing temperature and $H_2$/CO feed ratio led to a decrease in chain growth probabilities followed by increased methane and lower $C_{5+}$ selectivity. On the other hand, increasing the pressure had a positive but slight effect on $C_{5+}$ selectivity. Presence of $CO_2$ also plays a role in selectivity of products. This effect is more important for iron-based catalysts because of the $CO_2$ hydrogenation ability of these catalysts. The different selectivity of the iron catalyst in the presence of CO and $CO_2$ can be attributed either to a different H/C ratio on the catalyst surface or the lower stability of the iron carbide in $CO_2$ atmosphere [12]. Another important parameter is the space velocity. It has been proved that the selectivity of methane and olefins decreases in low space velocities; meanwhile this change in space velocity does not affect the selectivity towards paraffins [5]. Water is another parameter that affects the selectivity of FTS products. Water is produced in varying amounts depending on the reaction conditions during FT reaction. In the case of iron catalyst, presence of water and $CO_2$ can lead to re-oxidization of iron. Due to the tendency of iron catalyst towards water-gas-shift reaction, partial pressure of water is increased with the time-on-stream. The excessive amount of water results in undesired effects on the rate of reaction and products' selectivity [145]. Pendyala et al. [146] evaluated the effects of water on performance of Fe catalyst in FTS. They added water at different reaction temperatures using a continuously stirred tank re-actor (CSTR). Water co-feeding at low temperature (230 °C) decreased the CO conversion, but at higher temperature (270 °C), addition of water led to an increase in CO conversion. The reason for this behavior was, because at lower temperatures oxidation is favored over carburization of Fe catalyst. In the case of Co catalyst, water affects CO conversion, $CH_4$ and $C_{5+}$ selectivity. In fact, water influences the hydrogenation, dehydrogenation, methanation, chain initiation, chain propagation reactions. Moreover, the impact of water on activity and selectivity of Co catalyst depends on the supports, which influences the degree of metal–support interaction determines the effect of water on catalytic activity and selectivity [147]. Dalai and Davis. [148] reported that water vapor affected the reduction behavior of the various Co-supported catalysts. They showed that in the case of silica-supported Co catalysts, water effect led to an increase in CO conversion, whereas for alumina, this effect was negative. The effect of carbon nanofiber as a support for Co catalyst was found to be positive [149]. Related to the effect of water on the oxidation of the metallic Co particles, it has been suggested that there are two mechanisms for oxidation of Co particles, direct and indirect oxidation. While the first mechanism deals with oxidation of Co via an $H_2O$ splitting mechanism, the second route postulates that the formation of CoO species is due to the highly electronegative oxygen species (O*) generated by CO dissociation during FT reaction. In situ magnetic measurements have provided support for indirect oxidation mechanism of Co catalyst [150]. Small metal Co particles (<4.4 nm) are more exposed to oxidation in the presence of water [151]. Wang et al. [152] have indicated that particle size plays an important role in oxidation of Co catalyst by water. Accordingly, by using kinetic data, they have shown that the oxidation of small Co particles (1.4–2.5 nm) by water vapor was more evident than the larger Co particles (3.5–10.5 nm).

### 8.3. Effects of Reactor Design

Optimization of reactor design as well as operational conditions plays a pivotal role in controlling the product selectivity of FTS. The Fischer–Tropsch process is a highly exothermic reaction, so the heat transfer management is one of the main challenges in

reactor design. Without an effective heat removal, high methane yield, carbon deposition and catalyst deactivation will result [153]. The most employed commercial reactor designs in FTS can be classified into four groups: (a) slurry bubble column reactor (SBCR), (b) multi-tubular trickle-bed, (c) circulating fluidized-bed, and (d) fixed fluidized-bed reactors (Figure 6).

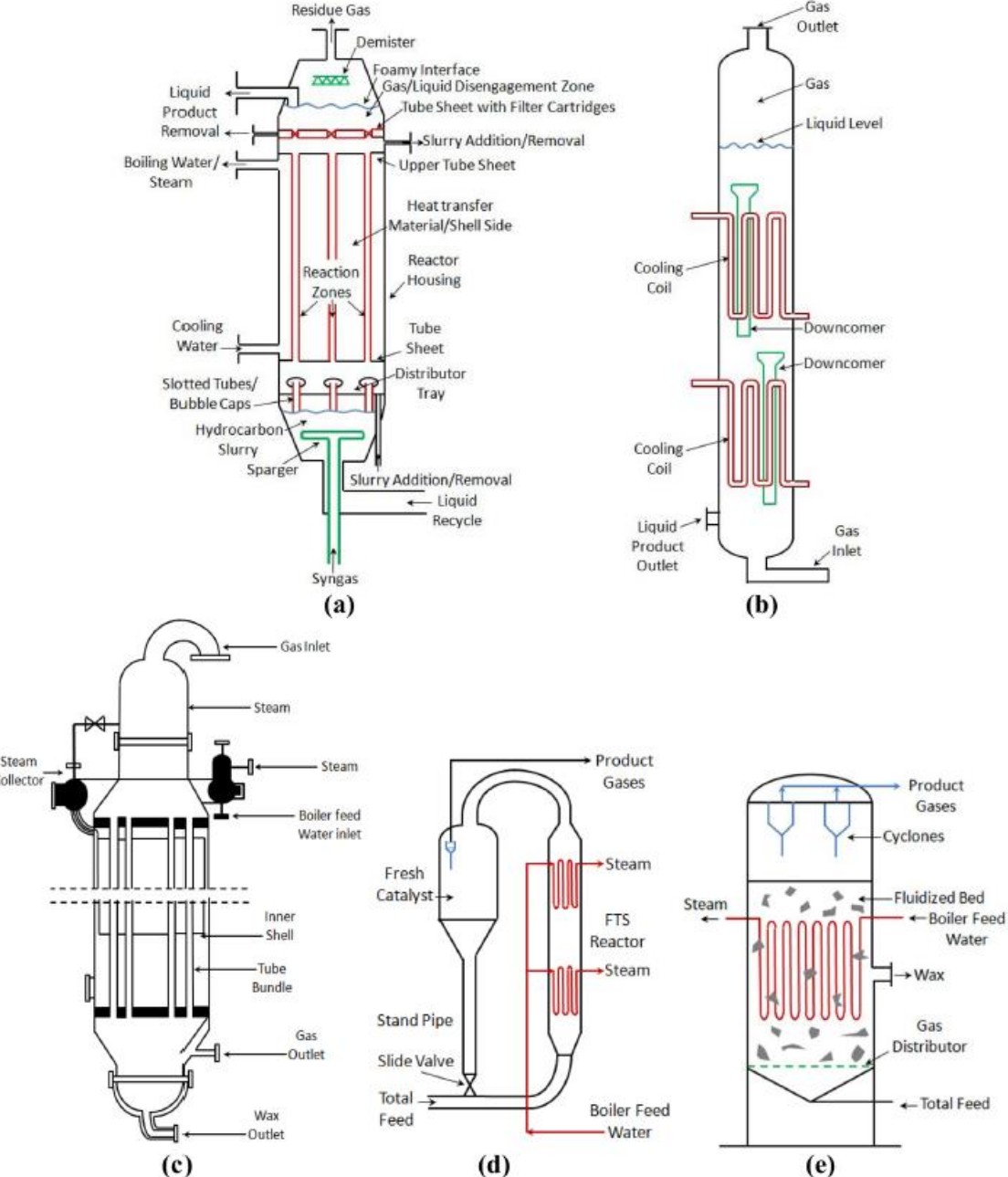

**Figure 6.** Schematic of commercial reactor designs for FTS. (**a**) Exxon and (**b**) Sasol slurry bubble column reactors (SBCRs), (**c**) multitubular trickle-bed reactor, (**d**) circulating fluidized-bed reactor, and (**e**) fixed fluidized-bed reactor. Reproduced with permission [12].

The SBC reactors provide efficient heat transfer and mass transfer and contain no mechanically moving parts. Thus, they have reduced wear and tear, higher catalyst durability, ease of operation, and low operating and maintenance costs [154]. These reactors provide more uniform temperature distribution compared to the fixed-bed reactor system. Moreover, the gas bubbles in the liquid phase help the catalyst particles to be in a dispersed

state. Commercially, middle distillates are more favored in SBC reactors [155–157]. Exxon and Sasol have developed SBCRs as shown in Figure 6a,b for FTS. These types of reactors have also some limitations such as back mixing of the gas phase, which decreases gas-liquid mass transfer rate, and consequently lowers conversion and selectivity of the FTS catalyst. Furthermore, catalyst separation from liquid products and difficulties for the scale-up impose additional limitations on the application of SBC reactors [157]. Fixed-bed reactors (Figure 6c) remain an attractive approach in reactor development of FTS because of the high catalyst loading/reactor volume which leads to the higher productivity/reactor volume and simplicity of scale-up from a single tube to a pilot plant [155]. Some limitations of fixed-bed reactors are diffusion-related problems within the catalyst particle and inconvenient heat transfer. In the case of employing fixed-bed reactors, production of linear waxes, which can be selectively hydrocracked to diesel, is more favored [156]. Circulating fluidized bed is another reactor configuration which was developed by Sasol for FTS (Figure 6d). High operation temperature for achieving higher conversions, energy requirement for circulation of the catalyst and pressure drop led to replacement of these reactors with fixed fluidized-bed reactors (Figure 6e). Fluidized-bed reactors were developed mainly for the high temperature gas phase HTFT reactions. Unlike SBC reactors, the ease of catalyst loading and replacement during the reaction can be considered as an advantage for the fluidized-bed reactors [157]. Recently, there is a trend in utilizing multi-tubular reactors instead of conventional reactors. Operation of these reactors leads to the higher catalyst activity due to the efficient heat removal, but construction of these configurations requires higher cost [158]. Honeycomb monolith reactors can also be an appropriate choice for LTFT reaction, which leads to low pressure drop, efficient optimum catalyst utilization, and highly efficient gas-liquid mass transfer [159]. Another reactor design for FTS is the use of membrane reactors, which can improve the catalytic activity of this process. A catalytic membrane reactor provides a defined reaction zone, and the reactants are forced through the membrane by applying a pressure gradient. The distribution of the feed through the membrane enables better temperature control leading to lower methane selectivity [160]. The unwanted effects of water on oxidation of the active phases in FTS can be addressed by employing membrane reactors by selectively recovering the water molecules from the reaction module. Therefore, the reaction equilibrium shifts to hydrocarbons production [161]. Bellal and Chibane. [162] investigated the catalytic performance of iron catalyst in a membrane reactor. The removal of $CO_2$ and $H_2O$ by membrane during FTS reaction altered the $H_2/CO$ ratio, which affected the products' selectivity. They found that by using water permselective membrane, the selectivity of $C_3$–$C_5$ olefin compounds increased, and by separating $CO_2$, the selectivity of paraffins was boosted.

## 9. Tuning Products' Selectivity by Zeolites

One of the promising alternatives for liquid fuels production is direct synthesis of these fuels by using a bifunctional catalyst in the FT process. The bifunctional catalyst possesses active sites for chain growth and acid sites for hydrocracking/isomerization processes simultaneously. The active sites for chain growth are provided from FT active metals (Fe, Co, Ni, and Ru), whereas the acid sites are shared from acidic supports such as zeolites [163]. Chain growth occurs by means of CO dissociation followed by C-H and C-C coupling. On the other hand, hydrocracking and isomerization consists of breaking C-C bonds to create lighter products and formation of branched hydrocarbons for the sake of increasing the octane number, respectively [164,165]. In bifunctional catalysts such as an FT active metal supported on zeolite, carbocations can be generated by migration of formed alkene to the acid sites of the zeolite with subsequent protonation by Brønsted acid sites. Carbocations are reactive species and can be involved in reactions such as skeletal isomerization, double bond shift and β-scission. This mechanism has been reported by several authors but for an exact explanation of the hydrocracking on a bifunctional catalyst, deeper studies are needed [163,166,167]. The schematic of the mentioned mechanism for hydrocracking/isomerization is outlined in Figure 7. Zeolites are crystalline microporous

aluminosilicates that consist of frameworks assembled from tetrahedral units with Si or Al cations located in the center and oxygen atoms at the corners of the framework [59]. Due to the shape-selective, uniform pore structure and acidic nature of zeolites, they have gained much attention as a support material for FTS. The shape-selective structure of the zeolites limits chain growth and favors the formation of lighter hydrocarbons. On the other hand, acidity of the zeolites plays an important role in catalyzing secondary cracking and isomerization reactions of primary products, which shifts the selectivity of FT products to gasoline ($C_{5-11}$), jet fuel ($C_{8-16}$) or diesel fuel ($C_{10-20}$) formation [129]. Therefore, direct synthesis of liquid fuels can be achieved by using zeolite-supported catalysts in FTS [131]. Peng et al. [127] used a bifunctional catalyst which consisted of Co nanoparticles and zeolite H-Y. They detected that by using Co/Na-meso-Y catalyst, the selectivity towards diesel fuel production reached its maximum (60%) compared to other catalysts supported on conventional support materials such as alumina and silica, and this was due to the hydrogenolysis behavior of zeolites. Sun et al. [168] scrutinized the feasibility of Raney Fe@HZSM-5 catalyst for gasoline production via FTS. They found an unexpected increase in $C_{5+}$ selectivity of the catalyst and this was due to the effect of HZSM-5 in improving the abundance of Hägg carbide phase in the catalyst. In fact, the hydrophilic nature of the HZSM-5 provided a less oxidizing chemical environment around the core followed by stable iron carbide phases. In another work by Li et al. [131], they reported a Co catalyst supported on Y-type zeolite for direct synthesis of liquid fuels. They found that by incorporating $Ce^{3+}$ and $La^{3+}$ into the $Y_{meso}$ because of ion-exchange, resulting catalysts exhibited higher selectivity of liquid fuels with high isoparaffin content and this was due to control of Brønsted acidity and porosity of zeolites. In fact, when zeolites are applied in FTS along with transition metals, they can act either as a support material for providing high surface area for dispersion of the active sites and aiding cracking of heavier hydrocarbons ($C_{21+}$) by means of their tunable Brønsted acidity and porosity.

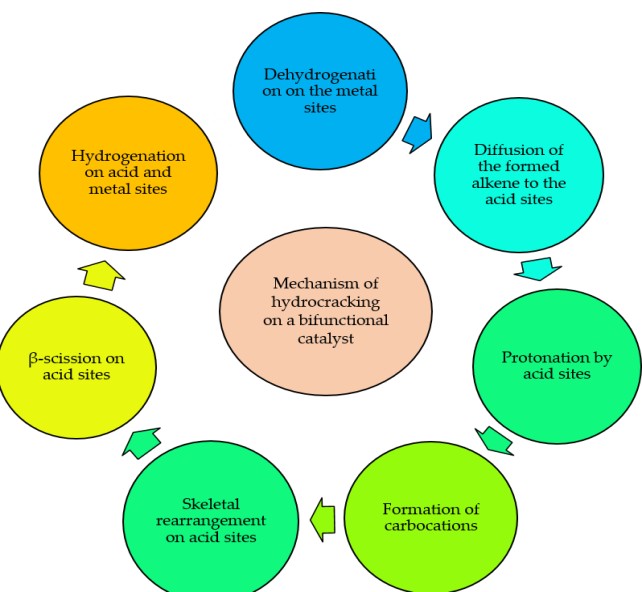

**Figure 7.** Schematic of hydrocracking steps on a bifunctional hydrogenation catalyst.

Liquid fuels produced by FTS are in high demand due to the lower amount of sulfur and aromatic compounds during their use, which decreases the air pollution. Liquid fuels can be directly synthesized in FTS reactor with no need for post-treatment and cracking by tunning operational conditions and employment of bifunctional catalysts. This also decreases the operational costs for GTL plants.

## 10. Deactivation of the Catalysts during FTS

The deactivation issue is one of the most challenging issues that should be addressed in catalytic reactions such as FTS. Deactivation can occur because of poisoning, fouling, thermal degradation, vapor compound formation accompanied by transport, vapor–solid and/or solid–solid reactions, or attrition/ crushing mechanisms [169]. Among different catalysts of FTS, deactivation of Fe-based FTS catalyst is more controversial due to its rapid deactivation under reaction conditions. Four main reasons for the deactivation of Fe-based catalysts are discussed here as follows [45,170–172]:

1.  The first possible mechanism is transformation of the active phases of Fe (iron carbides (such as χ-carbide, ε-carbide, ε'-carbide and metallic iron) to fewer active phases (magnetite and other types of iron carbides).
2.  The second postulated reason is the deposition of carbonaceous materials such as coke, graphitic and amorphous carbon. These materials decrease the effective contact between syngas molecules and the active sites of the catalyst for products formation.
3.  Sintering, the loss of catalytic surface area due to ripening or migration and coalescence of the iron phase under reaction conditions can be another reason for deactivation of iron catalysts.
4.  It is postulated that the sulfur compounds, which are present in most of the industrial syngas feeds, can cause the deactivation of the catalyst during FT reaction.

For Co catalysts, two main reasons of deactivation are carbon deposition and reoxidation of the Co particles. Furthermore, other possible reasons of deactivation for Co catalyst can be carbide formation, poisoning, surface reconstruction, the formation of stable compounds via interaction between cobalt and supports. The regeneration and rejuvenation of the FTS-based catalysts can be carried out by treatment of the catalyst with air (oxygen), hydrogen and/or CO for removing produced heavy waxes on the surface of the catalyst during reaction [173].

## 11. Conclusions

Fischer–Tropsch synthesis provides a promising route for converting non-fossil-based carbon-rich feedstocks to liquid fuels with almost zero amounts of sulfur, nitrogen, and aromatics. In this process, a wide range of hydrocarbons from methane to heavier waxes are produced simultaneously. The reaction mechanism of FTS has long been the subject of debate. A better understanding of the reaction mechanism leads to an effective catalyst design with high selectivity to the desired range of products. It has been accepted that instead of one single monomer, several monomer compounds take part competitively in the chain growth reaction. Quantum-chemistry and isotopic studies have significant effects on investigating the mechanism of FTS. The development of a specific scheme of reactions should involve the full range of FTS products and this makes the kinetic study of FT reaction a complex task. The kinetics models are developed for iron and cobalt-based catalysts ranging from simple power-law models to complex LHHW models by considering rigorous reaction mechanisms. This reaction proceeds over iron, cobalt, nickel, and ruthenium metals. Considering the high methane selectivity of nickel and scarce ruthenium resources, only cobalt and iron remain as suitable active metals for FTS. Supports are effective in dispersing active metals to improve the catalyst activity and selectivity to liquid hydrocarbons. The metal–support interaction is a crucial factor for reduction and dispersion behavior of the catalyst, which can be optimized by changing the physicochemical properties of the support. Although, $Al_2O_3$, $SiO_2$, $TiO_2$, $ZrO_2$, $CeO_2$ and carbonaceous materials have been widely used as support for the Co and Fe catalysts in FT synthesis, recently novel materials such as HAP have been getting attention as support of FTS catalyst. Plasma-spray method is a recent catalyst preparation technique, which involves a single step and is less labor-intensive as compared to impregnation and coprecipitation methods. Various characterization techniques such as spectroscopy, diffraction, microscopy, and thermal-based methods have been developed for analyzing physical, chemical, and electronical structure as well as reduction behavior of the FT

catalysts. In situ and operando characterizations using synchrotron have shed light on the mechanism and catalyst behavior of the FT process. The nature of active metal, support, operational conditions, and reactor configuration are important factors controlling the selectivity of FT products. On the other hand, applications of zeolites and bifunctional catalysts can tune the selectivity of FTS towards liquid fuels of better quality. Pore size and acidity of these support materials are two important factors that can be modified by utilizing larger pore zeolites and ion-exchange characteristics, respectively. By modifying the physicochemical properties of bifunctional catalysts and operational conditions, direct synthesis of high-quality liquid fuels can be achieved in GTL plants. Deactivation of the catalyst due to phase transformation, sintering, coke deposition and poisoning is one of the important challenges for FTS catalysts, which requires more studies in future.

**Funding:** This research was funded by Canada Research Chair (CRC) Program and Natural Sciences and Engineering Research Council of Canada (NSERC).

**Acknowledgments:** The authors acknowledge the funding from Canada Research Chair (CRC) Program and Natural Sciences and Engineering Research Council of Canada (NSERC) for this research.

**Conflicts of Interest:** The authors declare no conflict of interest.

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
