# Peer review of "Kinetics and Selectivity Study of Fischer–Tropsch Synthesis to C5+ Hydrocarbons: A Review"

_catalysts, doi:10.3390/catal11030330_

Round 1
Reviewer 1 Report
Line 12. GTL technology has never been based on Ru catalysts.
Line 50. The designation of olefins as (CH2)n is incorrect and could be confused with naphthenes.
Line 63. Boiling range of synthetic naphtha should be quoted, not a single boiling point.
Line 256. Not all suggested FTS mechanisms postulate the formation of surface RCHxOH species. In the cited review [48] these species are correctly assigned to one of the existing hypothesis.
Line 271, 298. In [52] no 14C tracers are mentioned. The remark about WWII is intriguing but hardly relevant.
Line 384. Products of FTS are not a mixture of liquid fuels but rather a mixture of the corresponding hydrocarbon fractions.
Line 385. The reason why hydrocracking units are included in modern GTL plants is not clear from the manuscript. In this respect I recommend citing the review [doi: 10.1016/j.mencom.2018.07.001] which addresses the issue in detail.
Line 396. ASF equation cannot predict properly the proportions of methane to gasoline etc. Methane is usually formed in excess over the value estimated from ASF equation, especially over Co catalysts.
Line 473. Simplicity of operation is not among the advantages of SBC reactor. On the contrary, gas phase back-mixing, catalyst separation, catalyst erosion and sophisticated reactor scaling-up are well-known issues associated with this type of reactors. Low molecular weight alkenes are NOT the most favored products in these reactors.
Line 483. Fluidized bed reactors are designed for HTFTS, it should be emphasized.
Throughout the text: don't forget subscripts in chemical formulas.
Reviewer 2 Report
Kinetics and selectivity study of Fischer-Tropsch synthesis to C5+ hydrocarbons. A review.
Authors: Z. Teimouri, N. Abatzoglou and A. K. Dalai.
Several reviews on the Fischer-Tropsch Synthesis have been published and it is not easy to present something new. Fe and Co are the commercial catalysts and the authors have tried to present both systems together. Fe and Co behave differently and they are therefore usually treated separately. The authors should get credit for this approach. I have only a few comments:
- The mechanism of FTS is a never ending story! Lately, H2 assisted CO dissociation on Co has received much support. The authors could have mentioned this.
- For Co water is a main product and water is important for the reaction. For Co catalysts the selectivity of C5+ increases with the CO conversion and it is believed that this could be due to water. Should have been included in the discussion.
- Co oxidation by water during FTS has been discussed heavily. It is possible that small Co particles could be oxidized by water, but not larger particles. The authors should have included some references to this discussion.
- On Co catalysts the formation of the carbide is not very likely in a system with large amounts of H2.
- The Co particle size is important for the activity and selectivity. H2 chemisorption is a usual way of determining the particle size, but is mot mentioned. What about the determination of particle sizes of Fe?
- It is written: Excessive amount of produced water, especially in the case of Fe as catalyst…Fe catalyst has high WGS activity?
- In situ studies at a synchrotron laboratory have shown to be a very powerful technique to examine FTS catalysts. It is mentioned, but may be it could be expanded.
The review can be published after a revision.
Reviewer 3 Report
By definition, a scientific review is a compilation of data from existing literary sources, united by the unique author's idea and leading to non-trivial conclusions. The first component in the manuscript under consideration is presented relatively well (156 references), the second is completely absent. Over the past two decades, there have been many reviews covering the most diverse aspects of Fischer–Tropsch synthesis – from the quantum chemistry of surface catalytic stages to the chemical technology of multiphase reaction processes – and the authors do not make any attempts to stand out from this variety. This fully applies to sections 2 to 9 (catalyst composition, mechanism, preparation of catalysts, deactivation, selectivity, role of zeolites, kinetics, respectively). Actually, the authors' lack of ideological content of their review is manifested even in the logic of constructing the sequence of these sections – the questions of mechanism and kinetics, as more general and fundamental, should be stated first, and the details of the behavior of catalysts depending on their composition, preparation method, the presence of zeolites, etc. etc. – should be stated further. It is obvious that only a brief mention of the fact that the selectivity of the process can be determined by the Anderson-Schultz-Flory distribution index and that any discussion of cases when this distribution is violated (including due to the use of zeolites) is complete absent call to the heavens. Under certain conditions, the chance for originality could be preserved due to section 10 (physicochemical studies of catalysts); it could be used as a basis for assessing the prospects for their development in the near future and approaching the ideal of a comprehensive description of the complex structure of modern Fischer–Tropsch catalysts. However, this section remains no more than a listing of literary sources dealing with one or another research methodology. I think this review should be rejected to improve.
Round 2
Reviewer 2 Report
The paper can be published as it is. The authors have answered my comments in a proper way
Author Response
Thank you so much for your valuable comments on our manuscript.
Reviewer 3 Report
The main positive impression is that the literature basis of the review have been expanded, the sequence of chapters has been made more logical, although further improvements are welcome. The main negative impression is that no original conclusions were drawn from all this data, not a single word in the "Conclusion" chapter was changed. The key requirement for its rewriting remains in force.
